# Computational drug repurposing strategy predicted peptide-based drugs that can potentially inhibit the interaction of SARS-CoV-2 spike protein with its target (humanACE2)

**Samuel Egieyeh**[1]*, **Elizabeth Egieyeh**[2⊙], **Sarel Malan**[3‡], **Alan Christofells**[4⊙], **Burtram Fielding**[5‡]

**1** Computational Pharmacology and Cheminformatics Research Group, Pharmacology and Clinical Pharmacy Unit, School of Pharmacy, University of the Western Cape, Cape Town, South Africa, **2** Pharmacology and Clinical Pharmacy Unit, School of Pharmacy, University of the Western Cape, Cape Town, South Africa, **3** Pharmaceutical Chemistry Unit, School of Pharmacy, University of the Western Cape, Cape Town, South Africa, **4** South African Medical Research Council Bioinformatics Unit, South African National Bioinformatics Institute, University of the Western Cape, Cape Town, South Africa, **5** Molecular Biology and Virology Research Laboratory, Department of Medical Biosciences, University of the Western Cape, Cape Town, South Africa

⊙ These authors contributed equally to this work.
‡ These authors also contributed equally to this work.
* segieyeh@uwc.ac.za

**Data Availability Statement:** All relevant data are within the manuscript and its Supporting Information files.

## Abstract

Drug repurposing for COVID-19 has several potential benefits including shorter development time, reduced costs and regulatory support for faster time to market for treatment that can alleviate the current pandemic. The current study used molecular docking, molecular dynamics and protein-protein interaction simulations to predict drugs from the Drug Bank that can bind to the SARS-CoV-2 spike protein interacting surface on the human angiotensin-converting enzyme 2 (hACE2) receptor. The study predicted a number of peptide-based drugs, including Sar9 Met (O2)11-Substance P and BV2, that might bind sufficiently to the hACE2 receptor to modulate the protein-protein interaction required for infection by the SARS-CoV-2 virus. Such drugs could be validated in vitro or in vivo as potential inhibitors of the interaction of SARS-CoV-2 spike protein with the human angiotensin-converting enzyme 2 (hACE2) in the airway. Exploration of the proposed and current pharmacological indications of the peptide drugs predicted as potential inhibitors of the interaction between the spike protein and hACE2 receptor revealed that some of the predicted peptide drugs have been investigated for the treatment of acute respiratory distress syndrome (ARDS), viral infection, inflammation and angioedema, and to stimulate the immune system, and potentiate antiviral agents against influenza virus. Furthermore, these predicted drug hits may be used as a basis to design new peptide or peptidomimetic drugs with better affinity and specificity for the hACE2 receptor that may prevent interaction between SARS-CoV-2 spike protein and hACE2 that is prerequisite to the infection by the SARS-CoV-2 virus.

**Funding:** The author(s) received no specific funding for this work.

**Competing interests:** The authors have declared that no competing interests exist.

## Introduction

Severe acute respiratory syndrome coronavirus-2 (SARS-CoV-2, also known as nCOV-2019), a virus that recently emerged as a human pathogen in the city of Wuhan in China's Hubei province, causes fever, severe respiratory illness, and pneumonia [1]. The World Health Organization (WHO) reported over 17 million confirmed cases globally, leading to at least 680,894 deaths as at 2$^{nd}$ August 2020 [2]. The pathogen was rapidly described as a new member of the genus betacoronavirus, closely related to many bat coronaviruses and the severe coronavirus acute respiratory syndrome (SARS-CoV) identified in 2003 [3]. The disease and the virus causing it were named Coronavirus Disease 2019 (COVID-19) and COVID-19 or the COVID-19 virus, respectively, by WHO [4]. Although many drugs and therapeutics, including hydroxy-chloroquine, azithromycin [5], remdesivir [6], and idasanutlin [7], are currently being tested or suggested, no effective prophylactics or therapeutics have yet been identified. This underscores the urgency to find the much needed intervention for this global pandemic.

Scientists are gaining insight into the biology of the virus, particularly how it binds with human respiratory cells through the human angiotensin-converting enzyme 2 (hACE2). The human angiotensin-converting enzyme 2 (hACE2) plays a role in cardiovascular and renal disease, diabetes and lung injury [8]. Interestingly, hACE2 has also been shown to be the receptor for human coronaviruses SARS-CoV, NL63-CoV and SARS-CoV-2 [9]. The hACE2 is a type I integral membrane glycoprotein orientated with the N-terminus and the catalytic site facing the extracellular space, where it can metabolize circulating peptides. The small C-terminal, cytoplasmic domain has a number of potential regulatory sites [10]. It was recently reported that the virus uses a tightly glycosylated spike (S) protein to pull into host lung cells via hACE2 [11]. The SARS-CoV S protein is a fusion protein with three subdomains consisting of a conserved core region, a variable or insertion loop and the C-terminal SD-1 domain. Binding of the SARS-CoV S protein to its receptor, hACE2 requires significant conformational changes in the S protein. The insertion loop of the S protein has been reported to contain the receptor-binding domain (RBD) [12]. This is similar to what has been recently reported for the S protein from SARS-CoV-2 [13]. Although the N-terminal domain (NTD) and a C-terminal domain (CTD) of the S protein from SARS-CoV-2 can function as a receptor-binding entity, nonetheless, similar receptor binding modes to hACE2 were observed for SARS-CoV-2-CTD and SARS-RBD [14]. The critical residues for receptor binding have been identified in the receptor-binding domain (RBD) of the SARS-CoV S protein and C-terminal SD-1 domain (CTD) of the SARS-CoV-2 S protein as well as in the interacting partner, hACE2 [15], thus making them targets for the discovery and development of vaccines and drugs for prevention and treatment of COVID-19 and other coronavirus infections.

In light of the urgent need for such vaccines and drugs for the COVID-19 pandemic, drug repurposing becomes an attractive strategy to find known and approved drugs for this new indication. This approach has gained impetus in medicine, especially in the search for a treatment for complex disorders and pandemics such as COVID-19 [16]. In this study, we conducted a virtual screening (HTVS) of 1070 compounds from the drug bank against interacting surface on hACE2 receptor to which the SARS-CoV-2 spike protein has been shown to bind to. The results showed significant binding of a number of approved drugs, experimental and investigational compounds to the target site. From the top 1070 drugs docked against the interacting surface of the hACE2 receptor, we described two peptide drugs with good predicted binding affinity and poses that might disrupt the protein-protein interactions between the SARS-CoV-2 spike protein and the hACE2 receptor. A number of the drugs predicted may be potentially repurposed for the prevention and treatment of COVID-19.

## Materials and method

### Compound library and preparation for molecular docking

The compound library consists of 1070 compounds (these were top ranked compounds chosen from an in house experiment of initial virtual screening of over 7000 compounds from DrugBank online against spike protein of Human *coronavirus HKU1* (HCoV-*HKU1*)). The selected compounds were imported into Molecular Operating Environment (MOE) database viewer. The compounds were prepared by washing (protonation at pH 7, explicit hydrogen added, 3D coordinates generated), adding partial charges and energy minimized with the Amber10: EHT force field, down to a RMS gradient of 0.05 kcal/mol/$\text{Å}^2$.

### Protein target and estimation of its average structure

The protein target was the hACE2 receptor extracted from SARS-CoV-2 spike protein-hCAE2 receptor complex. The crystal structure of receptor-binding domain of the spike (S) glycoprotein of SARS-CoV-2 bound to its receptor, hACE2 (6LZG) was downloaded from the RSCB protein data bank (PDB). We conducted a short molecular dynamics (MD) simulation to explore the conformational space of the initial model from PDB followed averaging of representatives from clusters of the conformers to obtain the average structure of the complex. The SARS-CoV-2 spike protein was deleted from the complex and the remaining hACE2 receptor was used for the docking study. The three-dimensional structure of hACE2 receptor was cleaned, minimized, solvated, and prepared for MD with AMBER forcefield (ff99SB) on the MDWeb server [17]. The simulation was done over 50 ns with Molecular Operating Environment (MOE) 2019.01 suite (*Molecular Operating Environment (MOE)* 2019.01). The average structure was viewed and compared to the downloaded structure using USCF Chimera [18].

### Molecular docking studies

The protein preparation module in Molecular Operating Environment MOE 2019.01(*Molecular Operating Environment (MOE)* 2019.01) was used to prepare the average structure hACE2 receptor for virtual screening. The binding site was defined as the interacting surface between the hACE2 receptor and the SARS-CoV-2 spike protein. Molecular docking simulation was carried out with the MOE Dock module using the induce-fit model [19]. The hACE2 receptor was set as 'Receptor'. The 'Triangle Matcher', which is suitable for standard and well-defined binding sites, was set as the ligand placement method. Thirty poses were generated for each ligand and the poses were scored according to the London ΔG scoring function. The thirty poses were taken through molecular mechanics (MM) refinement to get two final poses. The final docking score was evaluated with the GBVI/WSA ΔG scoring function with the Generalized Born solvation model (GBVI) [20]. UCSF Chimera [18] was used to visualize the poses.

### Molecular dynamics (MD) simulation of hACE2 receptor-ligand complex

The hACE2 receptor with and without a docked ligand (the ligand used here was Sar9 Met (O2)11-Substance P with a binding free energy of -10.63 kcal/mol) were subjected to molecular dynamic simulations using the Simulation module in Molecular Operating Environment MOE 2019.01(*Molecular Operating Environment (MOE)* 2019.01). The hACE2 receptor and hACE2 receptor-ligand complex were at different instances protonated and energy minimized with the MMFF94x force field to get the stable conformer of the protein complexes in a vacuum (molecular system). The molecular system was parameterized with "Optimized Potentials for Liquid Simulations" (OPLS-aa) forcefield, suitable for proteins and small organic molecules. Molecular dynamics simulations were done in three steps. We first heated the molecular

system to 310 K (37 $^{o}$C). Then an equilibration step was used to equilibrate the molecule system at 310 K (37 $^{o}$C) for 10 nanoseconds. Next, the simulation step was used to generate the trajectory of the molecular system at 310 K using the Nose–Poincare–Andersen (NPA) algorithm for 100 nanoseconds (time step of each simulation was set to 0.02 picoseconds). Visualizations and data analysis were performed with VMD software [21].

## Simulation and dynamics of SARS-CoV-2 spike protein-hACE2 receptor interaction

Protein-protein interactions (PPIs), especially between the SARS-CoV-2 spike protein and the human hACE2 receptor, play an important role in the establishment of the infection by the SARS-CoV-2 in humans. In this study, PatchDock [22], a free and open accessed web server, was used to predict and analyze the structures of PPIs between the spike protein and hACE2 in the presence and absence of one of the top-ranking ligands (BV2) from the docking studies. FireDock (Mashiach et al., 2008) was used to refine the output from PatchDock and tease out the various interaction energies involved in the simulated PPIs. Molecular visualization of the output was done with UCSF Chimera [18]. Following the prediction of the protein-protein interaction, the SARS-CoV2-hACE2 complexes with and without BV2 were subjected to molecular dynamic simulations using MOE 2019.01 software as described above.

## Results and discussion

### Average structure of hACE2 receptor

We conducted a molecular dynamics simulation on the crystal structure downloaded from the protein data bank, i.e. the SARS-CoV-2 spike protein-hACE2 complex, before separating hACE2 receptor from the complex for the molecular docking study. Following the molecular dynamics simulation, the results showed that the average root mean squared fluctuation (RMSF) per residue of the protein were between 0.30 Å to 1.33 Å (Fig 1). The RMSF per residue is the distance between a residue in the first frame (structure) of the protein under simulation and the same residue in the subsequent frames (structures) of the same protein. Higher RMSF values indicate greater flexibility during the MD simulation. The highest RMSF value was observed for Glutamic acid 69 located in the loop region closes to the interacting surface of hACE2 receptor. Five other residues in hACE2 and four residues in the SARS-CoV-2 spike protein, located mostly in the loop regions, showed higher fluctuation > 0.8 Å indicative of some flexibility. However, residues, which constitute the interacting surfaces of the loop region of the spike protein and the alpha helix region of the hACE2, showed lower flexibility (RMSF between 0.3 Å and 0.6 Å). Overall RMSF values were higher for a number of residues outside the interaction surface of the complex than those within the interaction surface. The average structure of the complex generated from the molecular dynamic simulations was compared to the downloaded protein data bank (PDB) crystal structure (inserted in Fig 1). The result showed a root mean square deviation of 0.622 Å between the two complex structures. The larger RMSF values indicate increased random motions for residues outside the interaction surface of the complex, and the converse for residues within the interaction surface of the complex. This is a pointer to the involvement of the latter set of residues in the interaction between the S protein and the hACE2 receptor. In addition, root mean square deviation of 0.622 Å was observed between the two complex structures These observations suggest potential conformational changes and moderate flexibility of the S protein and the hACE2 receptor complex. Hence the need to estimate the average structure of the hACE2 receptor before the molecular docking studies.

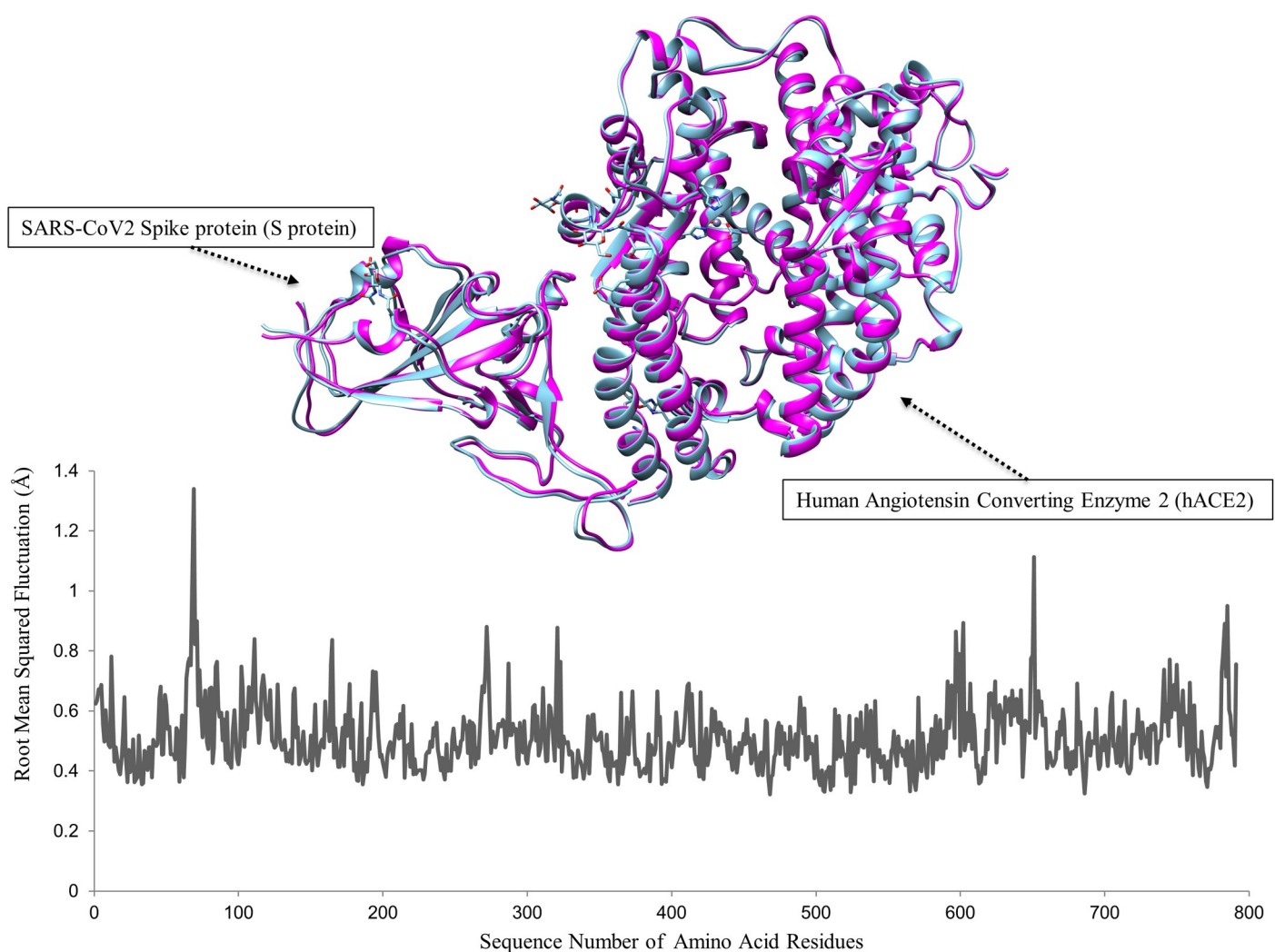

**Fig 1.** The root mean squared fluctuation (RMSF) per residue plot of the hACE2 receptor obtained from 50 ns molecular dynamics simulation in water at 37 °C (A). The RMSF per residue measures the average deviation of each protein residue over time from a reference position (i.e. the time-averaged displacement of the protein residues). Thus, RMSF analysis reveals the degree of fluctuation of portions of the protein. Inserted is the three-dimensional (3D) superimposition of the average structure of the SARS-CoV-2 S protein and hACE2 receptor complex generated from the molecular dynamics simulation (magenta) and the crystal structure of the SARS-CoV-2 spike protein and hACE2 complex downloaded from RSCB protein data bank (cyan blue) (B). An average root mean square deviation (RMSD) of 0.622 Å was observed for the complex.

## Molecular docking studies predicted peptide drugs with potential to bind to the interacting surface of hACE2 receptor

To predict currently approved drugs that will potentially bind to the interacting surface of the hACE2 receptor, compounds from Drug Bank were docked against this binding site using the induce-fit model in MOE 2019.01. An induced-fit docking model will capture the ensembles of the protein conformations which provide relatively high enrichment of the poses of the docked compounds. The initial London free energy of binding generated for 30 poses were rescored with generalized Born volume integral (GBVI) model to get the final S score for five poses. Higher negative S score is a predictor of better ligand interaction with the protein and more stable the ligand-protein complex. The chemical structures and estimated free energy of binding (S score) of the top poses of the compounds are presented in S1 Table. The top-

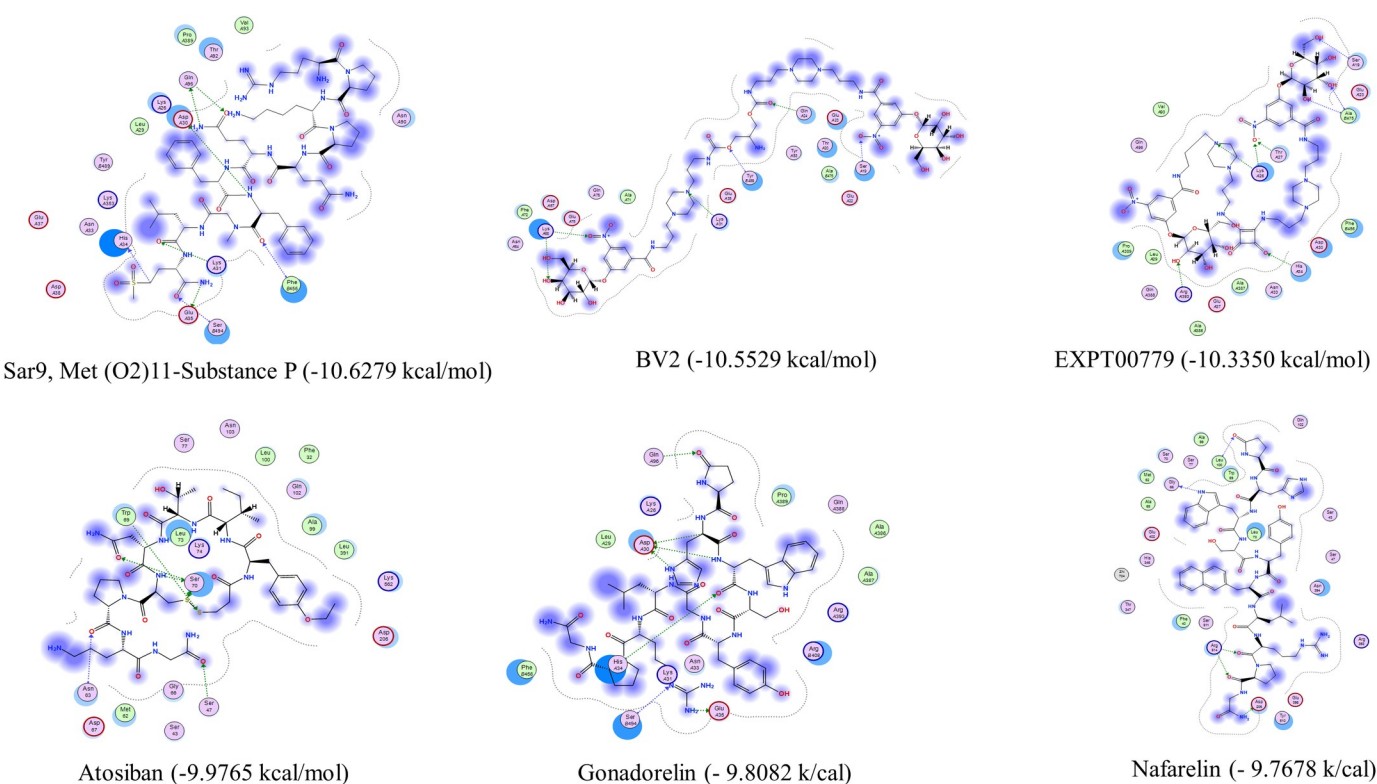

Sar9, Met (O2)11-Substance P (-10.6279 kcal/mol)

BV2 (-10.5529 kcal/mol)

EXPT00779 (-10.3350 kcal/mol)

Atosiban (-9.9765 kcal/mol)

Gonadorelin (- 9.8082 k/cal)

Nafarelin (- 9.7678 k/cal)

**Fig 2. A 2-dimensional visualization of the ligand-residue interactions of the top poses of six peptide drugs at the interaction surface of the hACE2 receptor.**

ranking compounds based on the predicted S score were mostly peptide-based drugs (approved, investigational and experimental drugs).

Ligand interactions of six top-ranked drug hits with the hACE2 receptor were generated (Fig 2). The number of hydrogen bond interactions of the top six ranked drug hits with residues within the interaction surface of the hACE2 receptor are shown as a heatmap in Fig 3A. The predicted interactions of peptide drug hits with these amino acid residues that are involved in the protein-protein interaction between the SARS-CoV-2 spike protein and its human target (hACE2 receptor) (Fig 3B) might disrupt the protein-protein interaction, thus limits the infectivity of the virus. Molecular docking of the drug library into the average structure of hACE2 receptor, which is derived from an average of the multiple conformations of the hACE2 receptor, provided us with an enrichment of docked compounds with reliable predicted binding energies/affinities. The molecular docking studies predicted drugs that can potentially bind strongly to the virus's spike protein interacting surface on the hACE2 receptor.

## Stability analysis of the one of the docked peptide drug-hACE2 receptor complex

We explored the stability of the predicted interaction between the top ranked peptide drug, Sar9 Met (O2)11-Substance P, and hACE2 receptor by conducting molecular dynamics of the system. The stability of the ligand-protein complex system in an aqueous solution was examined using the parameters RMSD (root mean square deviation) and number of hydrogen bonds following a 100 ns unconstrained simulation of the docked structure of hACE2 receptor (bound to Sar9 Met (O2)11-Substance P) and the hACE2 receptor alone.

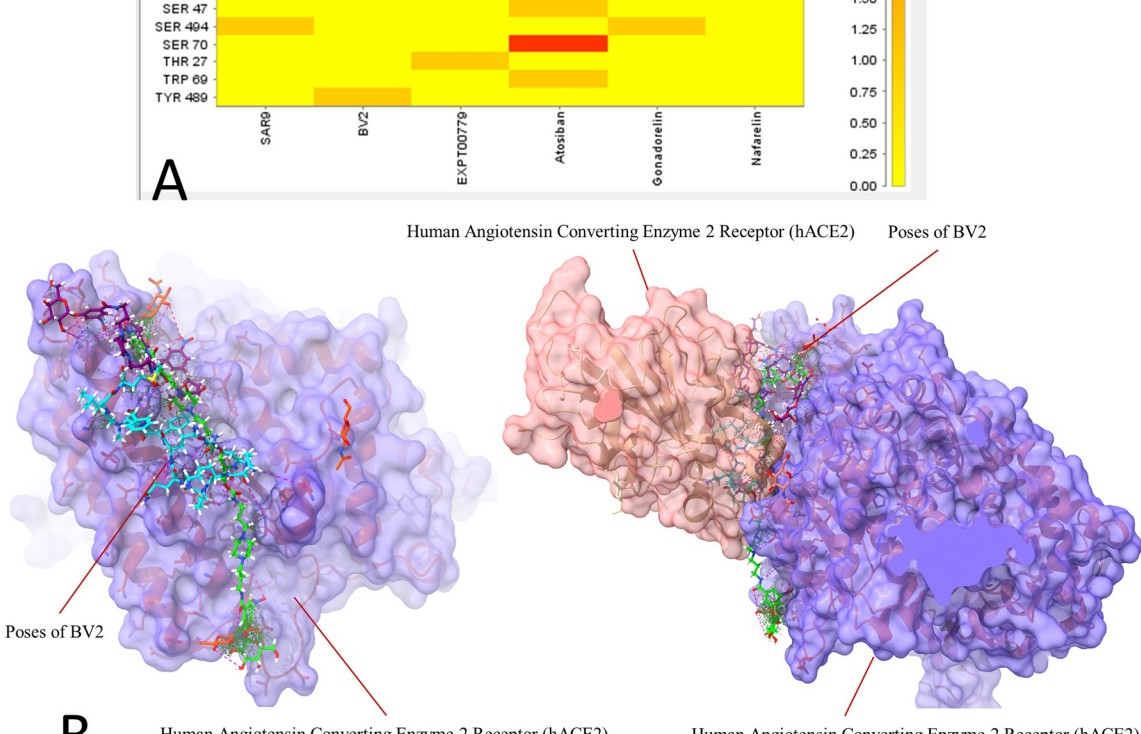

**Fig 3.** A heatmap of the count of the hydrogen bond interactions between the six top ranked peptide drugs and amino acid residues present at the interaction surface of the hACE2 receptor (A). Docked poses of one of the peptide drug hits (BV2) on hACE2 receptor that might interfere with the protein-protein interaction between the SARS-CoV-2 spike protein and its human target (hACE2 receptor) (B).

The overall average RSMD for the peptide drug, Sar9 Met (O2)11-Substance P, alone was 3.49 Å and it fluctuated between 3.1 Å and 3.8 Å during the 100 ns simulation. The high flexibility of this ligand alone, suggested by the RSMD, is expected due to the large number of rotatable bonds in the straight chain structure of this decapeptide. The average RSMD of the trajectories for the docked complex (peptide drug-hACE2 receptor complex) was 0.049 Å, which was slightly lower than the RSMD for the hACE2 receptor alone (0.059 Å). The RMSD of the backbone atoms of the bound and unbound hACE2 receptor showed relative stability during the simulation (Fig 4A) and the difference in the RSMD of the docked and undocked hACE2 receptor was 0.01 Å. This suggests that the configurational entropy of the hACE2 receptor backbone might decrease slightly upon binding of Sar9 Met (O2)11-Substance P, thus attesting to the potential stability of the docked Sar9 Met (O2)11-Substance P and the hACE2 receptor complex.

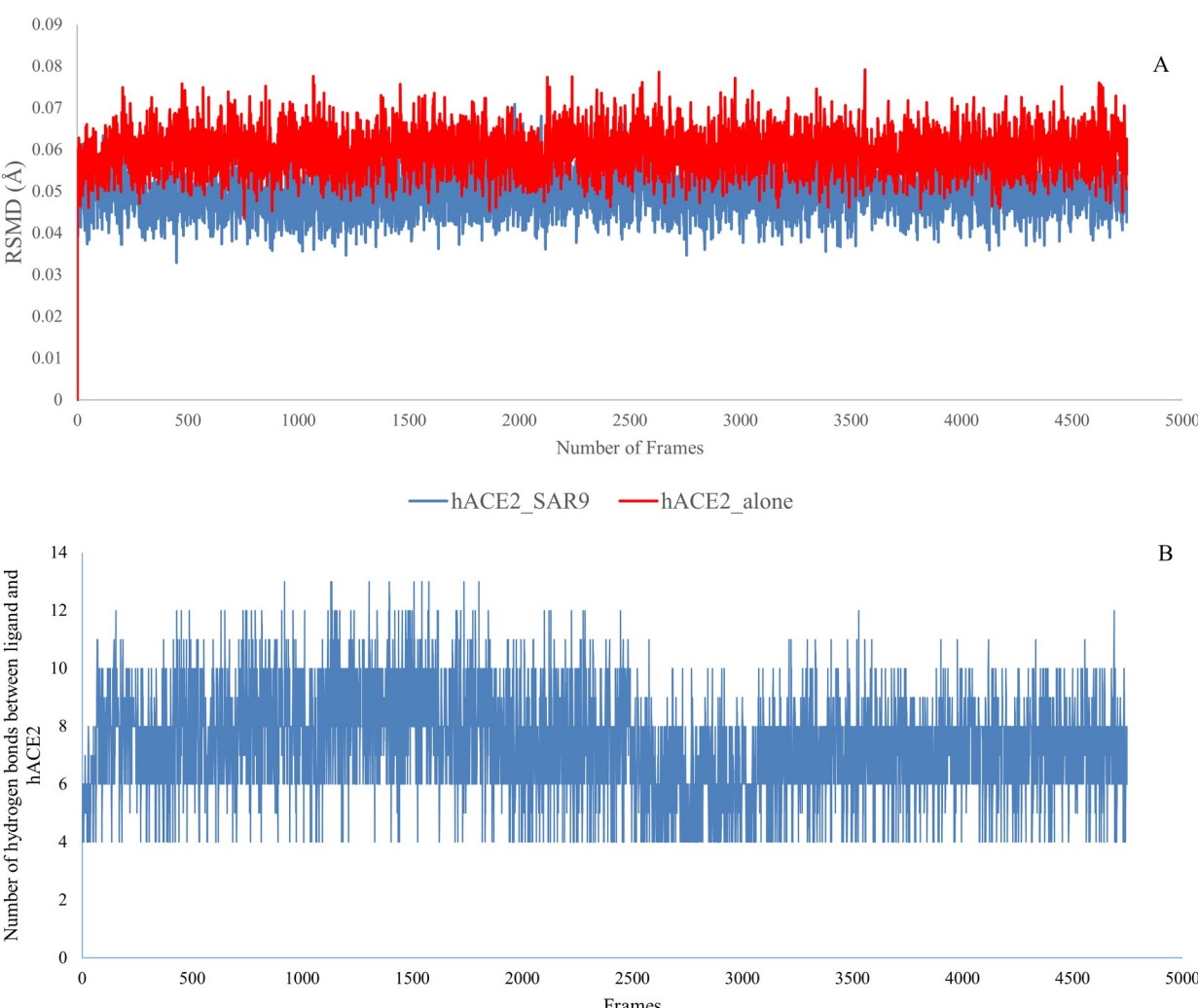

**Fig 4. Top figure shows the plot of RMSD (root mean square deviation) of the interacting surface of hACE2 receptor, in the presence (blue) and absence (red) of docked Sar9 Met (O2)11-Substance P, against the frames from the simulation.** This prediction suggests that there was no significant change in flexibility or conformation of the interacting surface of hACE2 upon binding of the peptide-drug. Bottom figure shows the plot of the number of hydrogen bonds between Sar9 Met (O2)11-Substance P and the hACE2 receptor across the frames from the simulation. The wide fluctuation of the number of hydrogen bonds between the docked ligand and hACE2 receptor might be due to the high flexibility of the docked ligand (a linear decapeptide).

Heteroatom-hydrogen bonds interactions play a critical role in the formation of protein-ligand complexes. Regarding stability of the hydrogen bonds (set at donor-acceptor distance of 3.0 Å and angle cutoff of $20^{\circ}$) between docked Sar9 Met (O2)11-Substance P and the hACE2 receptor, the number of hydrogen bonds fluctuated between a minimum of four and a maximum of thirteen during the 100 ns simulation (Fig 4B). The wide fluctuation of the number of hydrogen bonds between the docked ligand and hACE2 receptor might be due to the high flexibility of the docked ligand (as evidenced by the high RSMD value of 3.49 Å). This result revealed the need for a strategy like cyclization to reduce the entropy peptide-based inhibitors that may be designed for target protein, hACE2 receptor.

Overall, the root-mean-square deviation (RMSD) analyses of hACE2 backbone atoms and the ligand atomic coordinates as well as the dynamics of the number of hydrogen bonds were performed to measure the structure stability of the top ranked peptide drug-hACE2 receptor

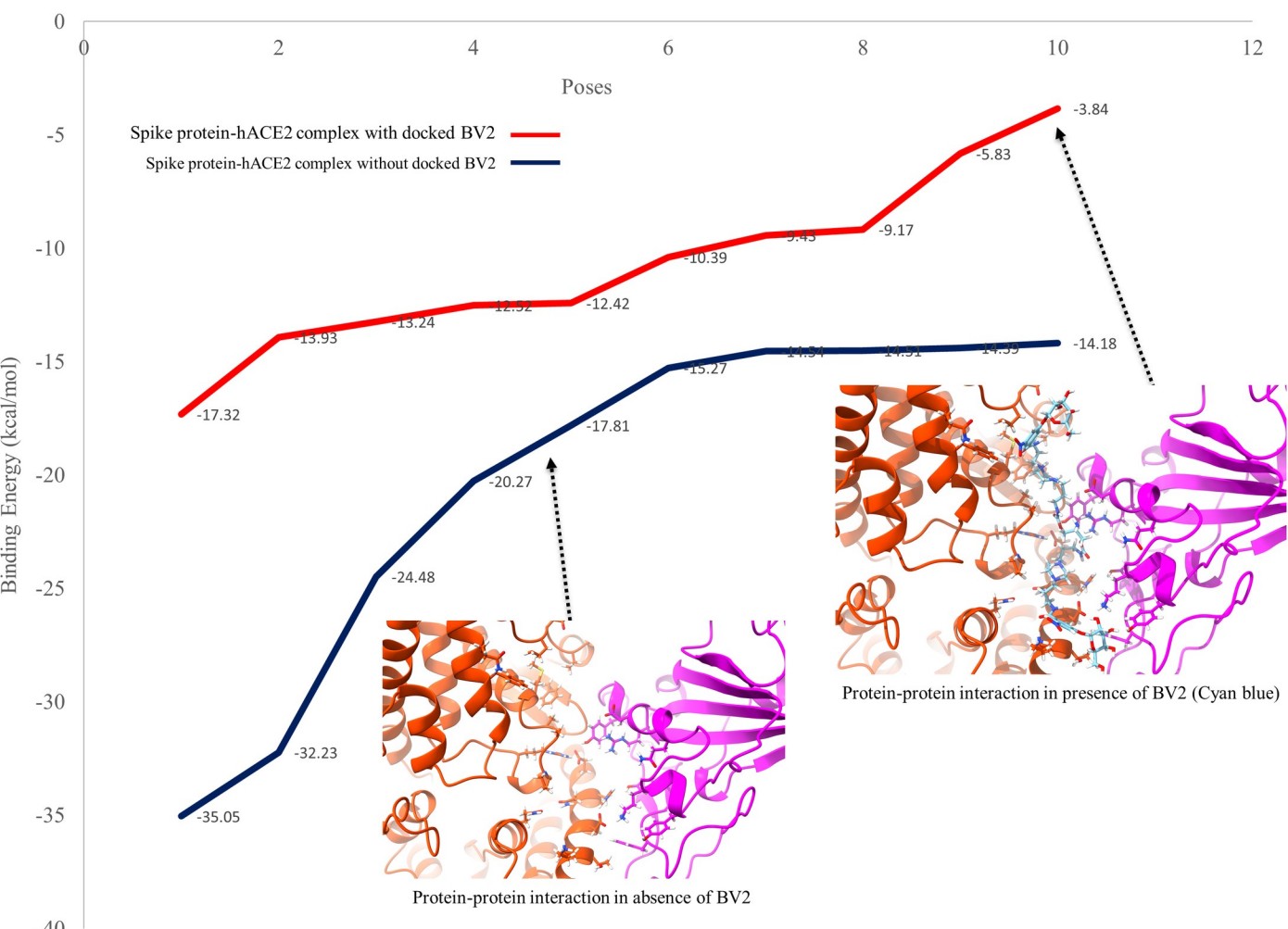

**Fig 5. A plot of the top 10 poses from the spike protein-hACE2 receptor interaction simulation in the presence of docked BV2 (red) and in the absence of docked BV2 (navy blue) on the x-axis and the binding free energies between the protein-protein interactions on the y-axis.**

complex. The results suggest that the predicted Sar9 Met (O2)11-Substance P-hACE2 receptor complex was relatively stable even though the number of hydrogen bonds might fluctuate.

## Potential modulation of protein-protein interaction between spike protein and hACE2 receptor by a docked peptide drug

We hypothesized that the top-ranked peptide drug hits predicted in the study might modulate the energy of protein-protein interaction between the SARS-CoV-2 spike protein and hACE2 receptor. We conducted a protein-protein docking study to analyze the nature of protein–protein interactions (PPIs) between SARS-CoV-2 spike protein and human ACE2 receptor in the absence and presence of a docked peptide drug hits (BV2). The predicted binding free energy for the top 10 poses from the protein-protein interaction simulation (in the presence and absence of BV2) were plotted against the respective binding free energies in Fig 5. The predicted binding free energy between spike protein and hACE2 receptor in the absence of a peptide drug hit was -35.05 kcal/mol for the topmost pose. However, the predicted binding free energy between spike protein and hACE2 receptor in the presence of a drug hit (BV2)

significantly reduced to -17.32 kcal/mol for the top pose. These predictions suggest that BV2 might limit the interaction between the SARS-CoV-2 spike protein and hACE2 receptor.

## Structural stability analysis of the predicted protein-protein interaction between the spike protein-hACE2 receptor

The goal here is to study the stability of the predicted protein-protein interaction between the SARS-CoV-2 spike protein and hACE2 receptor. We assessed the root mean square of fluctuation (RMSF) of the residues in the protein-protein interacting regions of the two proteins as well as the root mean squared deviation (RMSD) of the backbone of the protein in the same regions. Generally, the RMSF value of a residue represents the local flexibility of a protein. It reflected the mobility of atoms that make up that residue across the trajectories during the MD simulation. Therefore, higher RMSF values for residues indicate higher mobility (less stability) of such residues. Similarly, higher RMSD value is an indication of greater flexibility (less stability) of the protein backbone. The results for these analyses are presented in Fig 6. The image at the bottom (Fig 6A) is a plot of the sequence number of the amino acid residues against their RMSF values for the spike protein-hACE2 complex in the presence of the docked BV2 (red) and absence of the docked BV2 (navy blue).

It can be observed from this plot that the spike protein-hACE2 complex in the presence of the docked drug hit (red) showed significant increase in fluctuations of the residues across most regions of the complex. Particularly, the regions that constitute the interacting surface between the spike protein (sequence number 1–68) and the hACE2 receptor (sequence number 716–771) showed great difference in fluctuations of the residues in the presence and absence of the docked drug. This observation was mirrored by the difference in RMSD of the backbone of the interacting regions of the protein complex in the presence and absence of the docked drug hit (shown on the top left (B) and top right (C) of Fig 6). This indicates an increase in the flexibility of the backbone of the protein in the interacting regions in the presence of the docked drug hit. The increased fluctuations of residues and flexibility of the protein backbone observed in the interacting region suggest increased mobility and less stability of the residues in that region of the complex. This suggests that the docked drug disrupted the stability of the spike protein-hACE2 complex by steric and/or electrostatic interference with the interacting residues that were holding the protein-protein complex together.

One possible explanation for the higher root mean square fluctuation (RMSF) and higher root mean square deviation (RMSD) of the protein backbone observed in presence of a docked drug might be the disruption of the erstwhile stable interaction between the SARS-CoV-2 spike protein and the hACE2 receptor by the docked peptide drug. This suggests the hypothesis that the docked drug has the potential to interfere with the interaction of the SARS-CoV-2 spike protein with the hACE2 receptor thus reducing the infectivity of the coronavirus. We quantify the extent of this potential interference of the interaction of the coronavirus spike protein with the host receptor by conducting a protein-protein interaction simulation in the presence and absence of one of the top drugs predicted in this study. The results showed that there was up to 50% reduction in the binding affinity between the proteins in the presence of the drug. This suggests the potential of this top ranked peptide drug (BV2), predicted in this study, to affect the protein-protein interaction required for the infection of the host by the coronavirus (SARS-CoV-2).

The aim of the current study was to predict drugs from the Drug Bank that can bind to the interacting surface on the human angiotensin-converting enzyme 2 (hACE2) receptor to which the SARS-CoV-2 spike protein binds to establish its infection. Such drugs could be further studied, validated and repurposed as potential inhibitors of the interaction of coronavirus

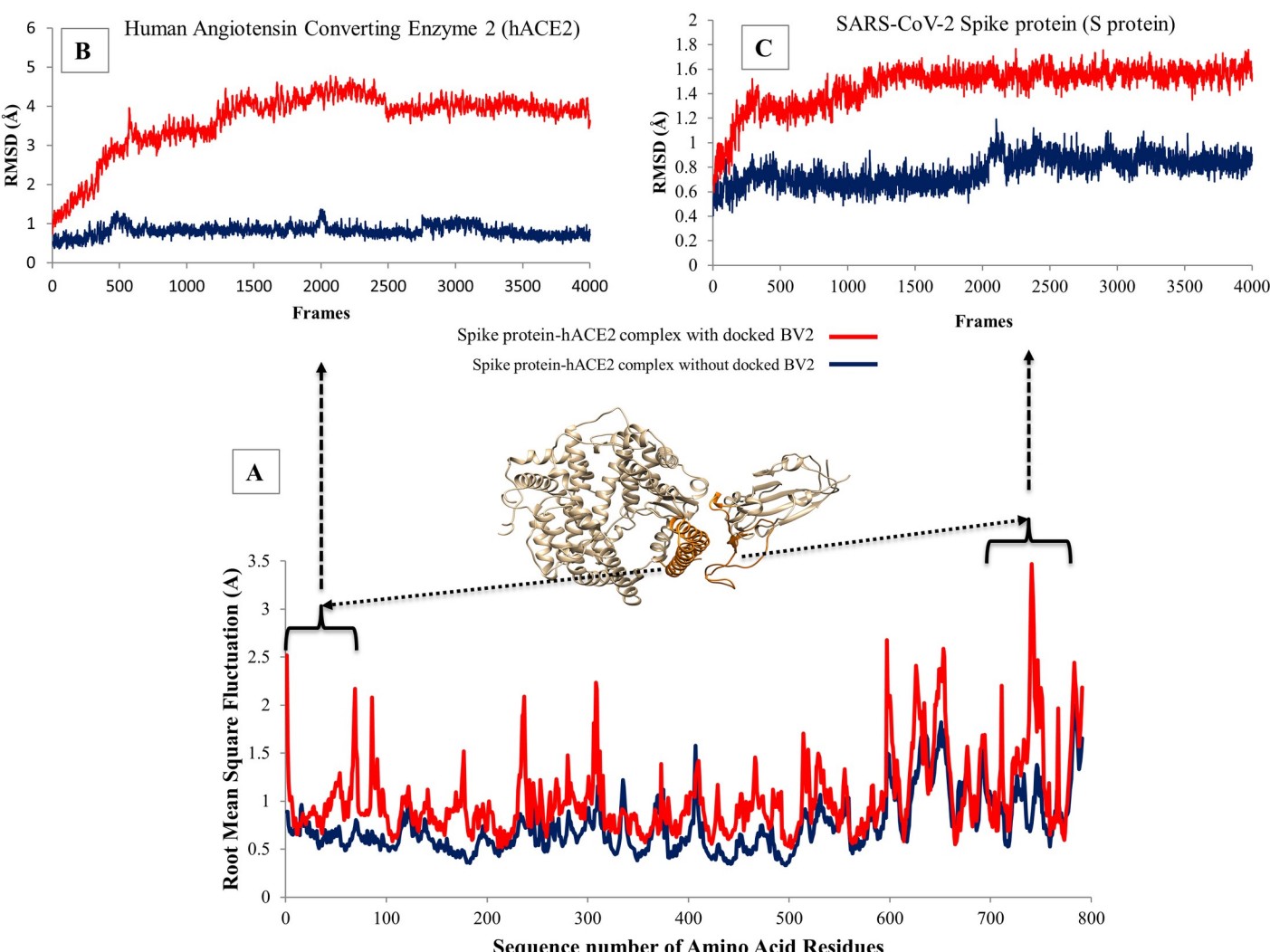

**Fig 6. Plots from the analysis of the molecular dynamic simulation of the predicted protein-protein interaction between SARS-CoV-2 spike protein-hACE2 receptor complex in the presence and absence of the docked BV2.** At the bottom (A), the plot of RMSF and the sequence number of amino acid residues of the spike protein-hACE2 complex in the presence (red) and absence (navy blue) of the docked BV2 is shown. The interaction surfaces are coloured orange in the model of the spike protein-hACE2 complex. Top left (B), shows the plot of RMSD of the interacting surface for hACE2 receptor, in the presence (red) and absence (navy blue) of BV2 within the complex, as a function of simulation time. Top right (C), shows the plot of RMSD of the interacting surface for SARS-CoV-2 spike protein, in the presence (red) and absence (navy blue) of BV2 within the complex, as a function of simulation time.

(SARS-CoV-2) with hACE2 receptors in the airway. In comparison to the "*de novo*" drug discovery process, drug repurposing considerably reduces the cost and time to bring new treatments to the market [23]. This is because the starting point for drug repurposing is the Drug Bank, which is a pool of approved and experimental drugs with well-documented toxicity, pharmacology, and drug-drug interactions [24]. In this study, a host of computational strategies and techniques were used to predict peptide drugs that may be potentially validated and repurposed for the treatment of COVID-19. Molecular docking studies have proven to be a very effective method for predicting ligand hits in structure-based drug discovery projects [25]. Molecular dynamics allows the visualization of the motions (flexibility) and stability of macromolecules involved in binding interactions [26].

**Table 1. The table shows a brief description of the preventive and therapeutic potential of two of the predicted top-ranked drugs against COVID-19.**

| S/N | Name | Binding Free Energy (S score, kcal/mol) | Current indication (s) and comments | Status and Pharmacokinetics |
|---|---|---|---|---|
| 1. | Sar9 Met (O2) 11-Substance P | -10.63 | An analog of the human neuropeptide Substance P that binds to neurokinin-1 receptor (a transmembrane protein which can be found throughout the body, including in the airways of humans and plays an important role in mediation of local and systemic inflammatory processes) [24]. It has been investigated for use/treatment in acute respiratory distress syndrome (ARDS) and viral infection [24]. | An investigational drug. Predictions using the SWISSADME server [28] revealed that it may have poor gastrointestinal absorption, no blood-brain barrier (BBB) penetration, be a P-gp substrate, and may not inhibit cytochrome P450 enzymes. Therefore, this peptide drug is expected to be administered intravenously. |
| 2. | BV2 (Cholera enterotoxin subunit B) | -10.55 | This is cholera enterotoxin subunit B. The GO term reported for this peptide (GO: 0046812) revealed that it is host cell surface binding drug. It interacts selectively and non-covalently with the surface of a host cell [29]. BV2 has been shown to bind up to five GM1 gangliosides present on the surface of the intestinal epithelial cells [24]. The foregoing attest to our observation that BV2 has the potential to bind effectively to hACE2 receptor and such may prevent protein-protein interaction required for infection by the COVID-19 virus. | An experimental drug. Although there are no pharmacokinetic data for this peptide [24], predictions using the SWISSADME server [28] revealed that it may have poor gastrointestinal absorption, no blood-brain barrier (BBB) penetration, be a P-gp substrate, and may not inhibit cytochrome P450 enzymes. Therefore, this peptide drug is expected to be administered intravenously, metabolized by peptidases in the blood and might have low distribution (concentrated within the vascular circulation) because it does not penetrate the BBB and may be effluxed by the P-gp. Prediction showed no tendency for toxicity. |

Their binding energy (S score) to the target site (hACE2 receptor), their current status in drug development and pharmacokinetics were also stated.

In all, we predicted that the docked peptide drug hits may provide steric and electrostatic hindrances to the protein-protein interaction of the SARS-CoV-2 spike protein and the hACE2 receptor via the C-terminal domain (CTD). We gained insight into the effect of the binding of one of the top-ranked peptide drugs (BV2) on the dynamics of the protein-protein interaction (PPI) between the SARS-CoV-2 spike protein and the hACE2 receptor by exploring the PPI with molecular dynamics. We also observed that all of the six top-ranked drug hits predicted in this study were peptides (biologics). The potential to inhibit protein-protein interactions (PPIs) has been demonstrated by the proliferation of biologics as therapeutic agents, especially in the treatment of autoimmune diseases [27]. The current indications, basic pharmacology relevant to COVID-19 and pharmacokinetics of two of the top-ranked peptide drugs predicted are described in Table 1.

## Conclusions

Conclusively, this study predicted peptide-based drugs that could potentially bind sufficiently to the interacting surface of the hACE2 receptor and could modulate the interaction of the SARS-CoV-2 spike protein with its host cells. Literature search revealed that the current or investigated indications for the predicted top-ranked peptide drugs were related to the pathophysiology or symptoms of COVID-19. Thus, the predicted top-ranked peptide drugs could be validated and repurposed to prevent and/or treat COVID-19 infection as well as be used as a template for the design of new peptidomimmetics against the virus. We hope that this drug repurposing hypothesis would stimulate further studies in the quest to find effective treatment for COVID-19.

## Supporting information

**S1 Table. The chemical structures and binding free energies of the five poses of the top 16 compounds from Drug Bank docked on to hACE2 receptor.**
(PDF)

## Author Contributions

**Conceptualization:** Samuel Egieyeh, Elizabeth Egieyeh.

**Data curation:** Samuel Egieyeh.

**Formal analysis:** Samuel Egieyeh.

**Methodology:** Samuel Egieyeh, Burtram Fielding.

**Supervision:** Alan Christofells, Burtram Fielding.

**Writing – original draft:** Samuel Egieyeh, Burtram Fielding.

**Writing – review & editing:** Elizabeth Egieyeh, Sarel Malan, Alan Christofells, Burtram Fielding.

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
