## [Decision Letter · Decision Letter 0]

28 Sep 2020

PONE-D-20-25309

Computational drug repurposing strategy identified peptide based drugs that can potentially inhibit the interaction of SARS-CoV-2 Spike proteins with its target (humanACE2)

PLOS ONE

Dear Dr. Egieyeh,

Thank you for submitting your manuscript to PLOS ONE. After careful consideration, we feel that it has merit but does not fully meet PLOS ONE’s publication criteria as it currently stands. Therefore, we invite you to submit a revised version of the manuscript that addresses the points raised during the review process.

In your revised manuscript please address the constructive comments of the two reviewers as fully as possible.

We look forward to receiving your revised manuscript.

Kind regards,

Israel Silman

Academic Editor

PLOS ONE

Journal Requirements:

Reviewers' comments:

Reviewer's Responses to Questions

**Comments to the Author**

1. Is the manuscript technically sound, and do the data support the conclusions?

Reviewer #1: No

Reviewer #2: Yes

2. Has the statistical analysis been performed appropriately and rigorously? 

Reviewer #1: N/A

Reviewer #2: N/A

3. Have the authors made all data underlying the findings in their manuscript fully available?

Reviewer #1: Yes

Reviewer #2: Yes

4. Is the manuscript presented in an intelligible fashion and written in standard English?

Reviewer #1: No

Reviewer #2: Yes

5. Review Comments to the Author

Reviewer #1: The aim of the study presented in this manuscript is to identify drugs from the Drug Bank that can prevent the binding of the SARS-CoV-2 spike protein with the human angiotensin-converting enzyme 2 (hACE2) receptor. However, it is difficult to understand that the authors used the whole S(RBD)-ACE2 complex, instead of the binding site of the S(RBD) protein or ACE2 alone, for molecular docking. In general, the screened compound or peptide is supposed to competitively binding to the S protein or ACE2. It is also quite curious how the binding of candidate peptides with the S(RBD)-ACE2 complex can destroy the stability of the complex. Does it mean that huge conformational changes were induced by the binding of the candidate or there was great conflict inside the molecular docked conformation? The simulation duration (100 ps or 4 ns) is too short to generate reliable information or conclusion. The definition of RMSF is inappropriate: “The RMSF per residue is the distance between a residue in the first frame (structure) of the protein under simulation and the same residue in the subsequent frames (structures) of the same protein.” Most importantly, there is no any in vitro experiment to validate the modeling result.

Reviewer #2: abstract - please change language to indicate all results are predictions, nothing was tested experimentally. for example: "The study identified a number of peptide-based drugs, including BV2, Gonadorelin,28 Icatibant, and Thymopentin that bind sufficiently" should be changed to "could potentially bind" or "is predicted to bind"

Please clarify how many drugs were screened. The methods section said 7000, line 80 said 1101 and line 83 said 1070.

Line 165 - "These observations suggest potential conformational changes and moderate flexibility of the S protein and the hACE2 receptor complex in vivo" - I would suggest removing the in vivo from the sentence.

Figure 1 - please indicate which color is the structure from PDB and which is the average of the simulation.

Line 178 "3.2 Docking studies identified drugs with good binding affinities to the target site" - since binding affinities were not measured, this should be changed to something like "3.2 Docking studies identified drugs with potential to bind to the target site"

Lines 184-186: “The more negative the S score, the stronger the binding affinity, the better the ligand interaction with the protein and the more stable the complex of the drug hits with the protein complex.” Again, language should indicate this is all predicted and not experimentally tested.

Figure 4 – please indicate what the red and blue lines are in the figure itself for better readability (it appears in the text). For the text and legend of figure 4, please indicate the drug name (it is only listed as “the top hit”).

Line 253 – “Top-ranked drug hits identified in the study could inhibit protein-protein interaction between the SARS-CoV-2 spike protein and hACE2.” This sentence is misleading, as it suggests experimental evidence of inhibiting the binding. Please delete it.

Lines 261-263: “Similar hinderance to protein–protein interaction necessary for the establishment of the infection by coronavirus was observed for all of the top six ranked drug hits.” – please provide the data, either in the text or a supplementary file.

Figure 5 – please indicate what the red and blue lines are in the figure itself for better readability (it appears in the text).

Line 280 – “from our previous study”. Please provide a reference

Lines 289-291: “The molecular docking studies identified drugs that bind strongly to the interacting surface of SARS-CoV-2 spike protein hACE2 complex.”. Please change to indicate these MAY bind strongly, as it is only predicted.

Lines 322-326: “Although a number of known drugs including hydroxychloroquine [5], Azithromycin [5], remdesivir [6], and idasanutlin [7] have been proposed and reported to be effective in the treatment of COVID-19, however results from our study suggest that hydroxychloroquine and remdesivir may have lower potential (as evidenced by binding free energy lower than -5.0 kcal/mol) to inhibit the interaction between the spike protein and hACE2.” There are several problems here. One – azithromycin and hydroxychloroquine have been widely shown to not be effective in treating COVID-19 patients, and should be removed from this sentence. Two – remdesivir is an RdRp inhibitor and is not expected to interfere with Spike binding to Ace2. Three- Ref #7 is a “thought experiment” is using idasanutlin to treat COVID-19, again in a mechanism unrelated to Spike-Ace2 binding. I suggest you delete these sentances.

In the Conclusion section (lines 327-337), please rephrase language to indicate potential inhibition rather than actual inhibition.

6. PLOS authors have the option to publish the peer review history of their article (what does this mean?). If published, this will include your full peer review and any attached files.

Reviewer #1: No

Reviewer #2: No

---

## [Author Response · Author response to Decision Letter 0]

12 Dec 2020

Following the review, which we accept in good faith, we have implemented the following changes according to the reviews (please note that number line of the previous version has changed in this new version): 

The title has been slightly revised to reflect that the study was a computational prediction. 

Reviewer #1: The aim of the study presented in this manuscript is to identify drugs from the Drug Bank that can prevent the binding of the SARS-CoV-2 spike protein with the human angiotensin-converting enzyme 2 (hACE2) receptor. 

However, it is difficult to understand that the authors used the whole S(RBD)-ACE2 complex, instead of the binding site of the S(RBD) protein or ACE2 alone, for molecular docking. In general, the screened compound or peptide is supposed to competitively binding to the S protein or ACE2. It is also quite curious how the binding of candidate peptides with the S(RBD)-ACE2 complex can destroy the stability of the complex. Does it mean that huge conformational changes were induced by the binding of the candidate or there was great conflict inside the molecular docked conformation? The simulation duration (100 ps or 4 ns) is too short to generate reliable information or conclusion. The definition of RMSF is inappropriate: “The RMSF per residue is the distance between a residue in the first frame (structure) of the protein under simulation and the same residue in the subsequent frames (structures) of the same protein.” Most importantly, there is no any in vitro experiment to validate the modeling result.

1. We revised the experiment and conducted the molecular docking on the hACE2 part of the S(RBD)-ACE2 complex. We have clarified that the molecular docking was done on the hACE2 (the spike protein interacting surface on hACE2) (Line 99 – 110)

2. Simulation to get the average structure of the protein was done over a duration of 50 ns. (line 107)

3. The definition of RMSF was revised (Line 156 – 158).

Reviewer #2: abstract - please change language to indicate all results are predictions, nothing was tested experimentally. for example: "The study identified a number of peptide-based drugs, including BV2, Gonadorelin,28 Icatibant, and Thymopentin that bind sufficiently" should be changed to "could potentially bind" or "is predicted to bind"

The statements were revised accordingly (Line 27 - 29).

Please clarify how many drugs were screened. The methods section said 7000, line 80 said 1101 and line 83 said 1070.

5. It has been clarified in the document that the amount compounds docked in this study was 1070. These compounds were the top subset from the drug bank database (over 7000 compounds) virtually screened against the spike protein of Human coronavirus HKU1 (HCoV-HKU1) in an inhouse study we conducted our lab (Line 92 – 94). 

Line 165 - "These observations suggest potential conformational changes and moderate flexibility of the S protein and the hACE2 receptor complex in vivo" - I would suggest removing the in vivo from the sentence.

The word “in vivo” has been removed.

Figure 1 - please indicate which color is the structure from PDB and which is the average of the simulation.

Correction effected: Inserted is the three-dimensional (3D) superimposition of the average structure of the SARS-CoV-2 S protein and hACE2 complex generated from the molecular dynamics simulation (magenta) and the crystal structure of the SARS-CoV-2 S protein and hACE2 complex downloaded from RSCB protein data bank (cyan blue) (B). (Line 182 – 185)

Line 178 "3.2 Docking studies identified drugs with good binding affinities to the target site" - since binding affinities were not measured, this should be changed to something like "3.2 Docking studies identified drugs with potential to bind to the target site" 

Corrected as suggested and rephrased (Line 188)

Lines 184-186: “The more negative the S score, the stronger the binding affinity, the better the ligand interaction with the protein and the more stable the complex of the drug hits with the protein complex.” Again, language should indicate this is all predicted and not experimentally tested.

It has been rephrased accordingly (Line 195 – 196)

Figure 4 – please indicate what the red and blue lines are in the figure itself for better readability (it appears in the text). For the text and legend of figure 4, please indicate the drug name (it is only listed as “the top hit”).

Corrected as directed. Legend inserted for the figure (now Figure 6) and top hit now specified (Line 305 – 314).

Line 253 – “Top-ranked drug hits identified in the study could inhibit protein-protein interaction between the SARS-CoV-2 spike protein and hACE2.” This sentence is misleading, as it suggests experimental evidence of inhibiting the binding. Please delete it.

Done. Deleted

Lines 261-263: “Similar hinderance to protein–protein interaction necessary for the establishment of the infection by coronavirus was observed for all of the top six ranked drug hits.” – please provide the data, either in the text or a supplementary file.

Statement has been deleted. Experiment was conducted on one predicted drug hit and presented as such.

Figure 5 – please indicate what the red and blue lines are in the figure itself for better readability (it appears in the text).

Done as suggested (see Figure 5).

Line 280 – “from our previous study”. Please provide a reference

Statement rephrased.

Lines 289-291: “The molecular docking studies identified drugs that bind strongly to the interacting surface of SARS-CoV-2 spike protein hACE2 complex.”. Please change to indicate these MAY bind strongly, as it is only predicted.

Statement rephrased to suggest that it was a prediction (340 – 342).

Lines 322-326: “Although a number of known drugs including hydroxychloroquine [5], Azithromycin [5], remdesivir [6], and idasanutlin [7] have been proposed and reported to be effective in the treatment of COVID-19, however results from our study suggest that hydroxychloroquine and remdesivir may have lower potential (as evidenced by binding free energy lower than -5.0 kcal/mol) to inhibit the interaction between the spike protein and hACE2.” There are several problems here. One – azithromycin and hydroxychloroquine have been widely shown to not be effective in treating COVID-19 patients, and should be removed from this sentence. Two – remdesivir is an RdRp inhibitor and is not expected to interfere with Spike binding to Ace2. Three- Ref #7 is a “thought experiment” is using idasanutlin to treat COVID-19, again in a mechanism unrelated to Spike-Ace2 binding. I suggest you delete these sentances.

Deleted as suggested.

In the Conclusion section (lines 327-337), please rephrase language to indicate potential inhibition rather than actual inhibition.

Done!

---

## [Decision Letter · Decision Letter 1]

28 Dec 2020

Computational drug repurposing strategy predicted peptide based drugs that can potentially inhibit the interaction of SARS-CoV-2 Spike proteins with its target (humanACE2)

PONE-D-20-25309R1

Dear Dr. Egieyeh,

We’re pleased to inform you that your manuscript has been judged scientifically suitable for publication and will be formally accepted for publication once it meets all outstanding technical requirements.

Kind regards,

Israel Silman

Academic Editor

PLOS ONE

Additional Editor Comments (optional):

Reviewers' comments:

Reviewer's Responses to Questions

**Comments to the Author**

1. If the authors have adequately addressed your comments raised in a previous round of review and you feel that this manuscript is now acceptable for publication, you may indicate that here to bypass the “Comments to the Author” section, enter your conflict of interest statement in the “Confidential to Editor” section, and submit your "Accept" recommendation.

Reviewer #1: All comments have been addressed

Reviewer #2: All comments have been addressed

2. Is the manuscript technically sound, and do the data support the conclusions?

Reviewer #1: Yes

Reviewer #2: Yes

3. Has the statistical analysis been performed appropriately and rigorously? 

Reviewer #1: N/A

Reviewer #2: N/A

4. Have the authors made all data underlying the findings in their manuscript fully available?

Reviewer #1: Yes

Reviewer #2: Yes

5. Is the manuscript presented in an intelligible fashion and written in standard English?

Reviewer #1: Yes

Reviewer #2: Yes

6. Review Comments to the Author

Reviewer #1: The authors have basically addressed the comments. It is thus recommended to accept this manuscript for publication.

Reviewer #2: (No Response)

7. PLOS authors have the option to publish the peer review history of their article (what does this mean?). If published, this will include your full peer review and any attached files.

Reviewer #1: No

Reviewer #2: No

---

## [Editor Report · Acceptance letter]

30 Dec 2020

PONE-D-20-25309R1 

Computational drug repurposing strategy predicted peptide-based drugs that can potentially inhibit the interaction of SARS-CoV-2 spike protein with its target (humanACE2) 

Dear Dr. Egieyeh:

I'm pleased to inform you that your manuscript has been deemed suitable for publication in PLOS ONE. Congratulations! Your manuscript is now with our production department. 

Kind regards, 

on behalf of

Prof. Israel Silman 

Academic Editor

PLOS ONE